# StreamFlow: Streamlined Multi-Frame Optical Flow Estimation for Video Sequences

**Shangkun Sun**
SECE, Peking University
Peng Cheng Laboratory
sunshk@stu.pku.edu.cn

**Jiaming Liu**
Tiamat AI
james.liu.n1@gmail.com

**Huaxia Li**
Xiaohongshu Inc.
lihx0610@gmail.com

**Guoqing Liu**
Minieye Inc.
liugq@ntu.edu.sg

**Thomas H Li**
SECE, Peking University
thomas@pku.edu.cn

**Wei Gao** ✉
SECE, Peking University
Peng Cheng Laboratory
gaowei262@pku.edu.cn

## Abstract

Prior multi-frame optical flow methods typically estimate flow repeatedly in a pairwise manner, leading to significant computational redundancy. To mitigate this, we implement a Streamlined In-batch Multi-frame (SIM) pipeline, specifically tailored to video inputs to minimize redundant calculations. It enables the simultaneous prediction of successive unidirectional flows in a single forward pass, boosting processing speed by $44.43\%$ and reaching efficiencies on par with two-frame networks. Moreover, we investigate various spatiotemporal modeling methods for optical flow estimation within this pipeline. Notably, we propose a simple yet highly effective parameter-efficient Integrative spatiotemporal Coherence (ISC) modeling method, alongside a lightweight Global Temporal Regressor (GTR) to harness temporal cues. The proposed ISC and GTR bring powerful spatiotemporal modeling capabilities and significantly enhance accuracy, including in occluded areas, while adding modest computations to the SIM pipeline. Compared to the baseline, our approach, StreamFlow, achieves performance enhancements of $15.45\%$ and $11.37\%$ on the Sintel clean and final test sets respectively, with gains of $15.53\%$ and $10.77\%$ on occluded regions and only a $1.11\%$ rise in latency. Furthermore, StreamFlow exhibits state-of-the-art cross-dataset testing results on Sintel and KITTI, demonstrating its robust cross-domain generalization capabilities. The code is available here.

## 1 Introduction

Optical flow estimation, which aims to model the per-pixel correspondence between two consecutive frames, is a fundamental task in computer vision. It has various downstream video applications, such as video generation [22, 33, 20], video editing [6, 51], and video compression [25, 19]. In video streams, there is frequently a need for multi-frame input and continuous optical flow estimation.

---

✉Corresponding author.

38th Conference on Neural Information Processing Systems (NeurIPS 2024).

Nevertheless, previous multi-frame methods usually perform spatiotemporal modeling in a pairwise way, which leads to redundant computation, as depicted in Fig. 1 and Alg. 1. Given input frames $I_{t-1}, I_t, I_{t+1}$, previous pairwise methods such as VideoFlow [38] only output *one* forward flow $F_{t\to t+1}$. To derive the consecutive flow $F_{t-1\to t}$, one additional forward pass with the input of $I_{t-2}, I_{t-1}, I_t$ is needed.

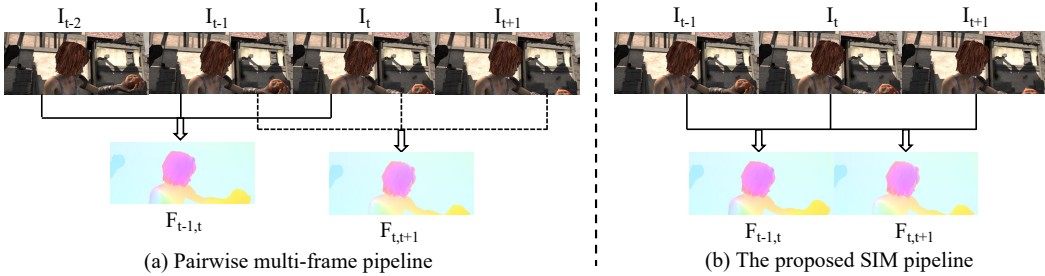

(a) Pairwise multi-frame pipeline     (b) The proposed SIM pipeline

Figure 1: Comparison between the pairwise and the proposed Streamlined In-batch Multi-frame (SIM) pipeline. Short dashed lines represent additional computations.

This gives rise to the design of what we refer to as in-batch estimation —a pipeline that simultaneously predicts all successive unidirectional flows within *one* forward pass, which greatly reduces the redundant computation. For instance, given input frames $I_{t-1}, I_t, I_{t+1}$, all consecutive flows $F_{t-1\to t}$ and $F_{t\to t+1}$ could be derived within *one* forward pass. It directly saves one forward pass time in the decoder. When $T$ frames are input simultaneously in the batch, there is still only *one* forward pass needed to obtain $T-1$ flows, and more time is served on average as $T$ increases.

Nevertheless, despite the acceleration brought by the in-batch estimation, its alteration of the pipeline renders it unable to directly apply past pairwise spatiotemporal modeling methods [16, 4, 38]. Notably, an effective spatiotemporal modeling approach is the key to resolving problems such as occlusions for multi-frame methods. Thus, this raises a new key question: *Under the constraint of in-batch estimation, how to perform effective spatiotemporal modeling while maintaining efficiency?*

In this work, we propose StreamFlow, a streamlined multi-frame optical flow estimation method tailored for video inputs. StreamFlow is made efficient through the Streamlined In-batch Multi-frame (SIM) pipeline, which avoids repetitive calculations when predicting unidirectional flows for videos. Furthermore, StreamFlow explores the challenge of effectively modeling spatiotemporal cues under the constraint of non-overlapping in-batch estimation. StreamFlow explores various spatiotemporal modeling methods and ultimately derives simple yet highly effective methods: a parameter-efficient Integrative spatiotemporal Coherence (ISC) module during encoding, and a lightweight Global Temporal Regressor (GTR) to decode all flows. Benefiting from these modules, StreamFlow achieves remarkable performance on Sintel and KITTI datasets *without self-supervised pre-training and the aim of bidirectional flows*. Moreover, StreamFlow attains state-of-the-art cross-dataset generalization results with comparable efficiency compared to two-frame methods, as illustrated in Figure 2.

In summary, our contributions are as follows: (1) We propose a Streamlined In-batch Multi-frame (SIM) pipeline for optical flow estimation, which eliminates the repetitive overlapping computation when computing forward flows for video inputs. (2) Under the constraint of a non-overlapping pipeline, we specifically designed the Integrative spatiotemporal Coherence (ISC) method to perform spatiotemporal modeling without additional parameters. (3) For the SIM pipeline, we devise a Global Temporal Regressor (GTR) during decoding to further exploit temporal cues with modest additional computation cost. (4) The proposed StreamFlow achieves superior performances on multiple benchmarks, particularly in occluded regions with comparable efficiency compared with two-frame methods, resulting in substantial improvements in optical flow estimation.

## 2 Related work

**Two-frame optical flow.** Optical flow estimation in the form of a supervised learning task has been performed by FlowNet [8] using Convolutional Neural Networks (CNN). The encoder-decoder architecture of FlowNet predicts flow from coarse to fine using the hierarchy of the flow pyramid. Thereafter, a number of refined coarse-to-fine approaches [12, 42, 43, 10, 11, 50, 54, 13] emerged.

The flow pyramid is constructed for the coarse-to-fine approach, which predicts the flow based on the flow guidance at a higher pyramid level. However, the flow guidance is often too coarse to capture small motions delicately and creates errors in later estimation. RAFT [45] recently introduced an iterative all-pairs flow transform technique, which enables the prediction of high-resolution flow and recurrent refinement of the residual flow estimation. It addresses the challenges of small motions and has consequently received high interest and performance in the field, inspiring numerous follow-up works [15, 28, 55, 21, 27, 56, 47]. To further address the occlusion issue, SKFlow [44] begins by expanding the spatial receptive field and designing effective large convolution kernel modules in the decoder of the flow network, with modest computational cost. Although StreamFlow is based on the SKFlow framework, it is designed to explore additional temporal cues and minimize redundant computations that are prevalent in earlier multi-frame methods. Their methodologies and foundational principles differ significantly.

**Multi-frame optical flow.** Exploiting temporal cues in optical flow estimation is an effective way to recover the occluded motion. Previous works [37, 46, 32, 1, 16, 4, 38, 26] propose various approaches to fuse temporal cues, such as leveraging previously predicted motion feature, optical flow, or contextual information. For instance, ContinualFlow [32] uses previous flow priors to estimate the current occlusion map. STaRFlow [1] passes extracted features in multiple scales, jointly with occlusion maps. [45] proposes a warm-start strategy to initialize the original flow with the past flow before prediction. MFCFlow [4] and MFRFlow [16] propose to leverage previously estimated motion features during decoding via feed-forward CNNs and self-similarity modeling, respectively. Nevertheless, these methods obtain a pairwise strategy when handling video sequences, which divide

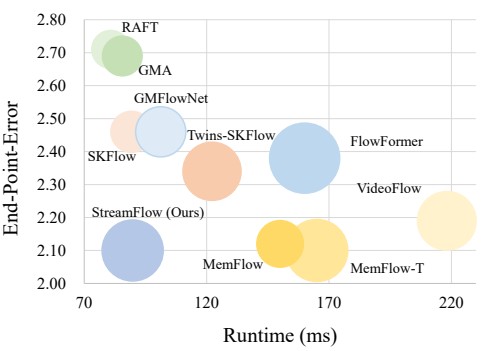

Figure 2: Comparison between performance and efficiency. A larger bubble denotes more parameters. Models are trained via the (C+)T schedule and tested on the Sintel final pass.

the input sequence into lots of overlapping groups and take huge repeated computations. Splat-Flow [46] utilizes the differentiable splatting transformation to explore temporal cues. TransFlow [26] decodes all flows simultaneously and achieves impressive results. However, it needs self-supervised pre-training on the flow datasets to help the temporal modeling modules converge. Besides, its pure transformer architecture does not have advantages in time. VideoFlow [38] employs TROF and MOP modules to utilize multi-frame temporal cues and bidirectional optical flow to effectively mitigate occlusion issues. Nevertheless, it still follows the pairwise method to predict multiple unidirectional flows with the cost of predicting bidirectional flows. Differently, StreamFlow is proposed to avoid redundant, overlapping computation for consecutive unidirectional flow predictions while exploring efficient and effective temporal modules design under such a pipeline. It addresses the redundancy problems previous pairwise methods including VideoFlow encountered, and achieves excellent accuracy with latency similar to some two-frame methods.

## 3 Methodology

In this Section, we introduce StreamFlow, an efficient and effective in-batch framework for multi-frame optical flow estimation. Extensive experiments guided our specific design of each module in StreamFlow. The key components of StreamFlow consist of three parts: (1) The Streamlined In-batch Multi-frame (SIM) pipeline for efficient multi-frame estimation, which contributes to the speed improvement of StreamFlow. (2) Integrative spatiotem-

---

**Algorithm 1** Pairwise Multi-frame Estimation

**Input:** frames $I_i$, size $N$, group size $T$
**for** $i = 1$ **to** $N - T + 1$ **do**
    $F_i =$ Model$(I_i, I_{i+1}, ..., I_{i+T-1})$.
**end for**

---

**Algorithm 2** StreamFlow Multi-frame Estimation

**Input:** frames $I_i$, size $N$, group size $T$
Initialize $i = 1$, stride= $T - 1$.
**repeat**
    $F_i, ..., F_{i+stride-1} = Model(I_i, ..., I_{i+stride})$
    $i = i + stride$
**until** $i + stride > N$
$j = N - stride$
$F_j, F_{j+1}, ..., F_N = Model(I_j, I_{j+1}, ..., I_N)$.

poral Coherence (ISC) modeling, which is parameter-efficient and is specifically designed for spatiotemporal modeling in the encoder. (3) Global Temporal Regressor (GTR), which is quite lightweight and learns temporal relations during decoding. We will first give an overview of our methods in Section 3.1, and then introduce each module in Section 3.2, Section 3.3, and 3.4, respectively. In the end, we discuss the supervision in Section 3.5.

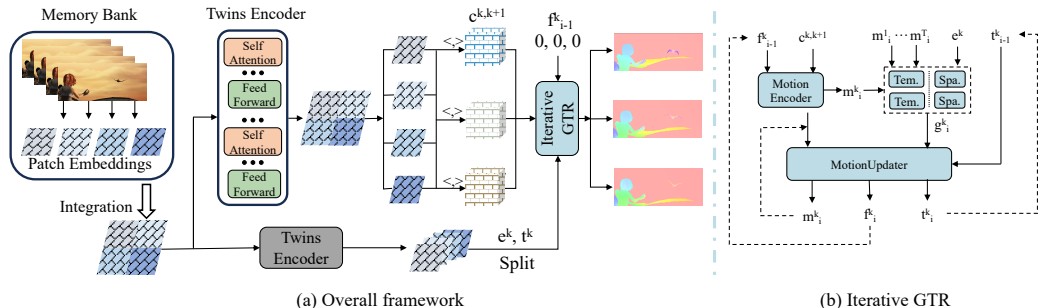

Figure 3: Overview of StreamFlow. (a) illustrates the overall framework and $<,>$ denotes the dot-product operation. The computation of cost volume is limited to adjacent frames and is performed once in one forward pass. Flows are initialized to zeros. (b) depicts the details of the GTR decoder.

## 3.1 Overview

The overall framework of StreamFlow is illustrated in Figure 3. For the basic encoder and decoder, similar to VideoFlow [38], StreamFlow adopts the Twins transformer [5] as the encoder and utilizes the motion encoder and updater in SKFlow [44] during decoding. The overall iterative-refinement design that adopts an iterative decoder is the paradigm proposed in IRR [12] and followed by a lot of subsequent works [44, 41, 15, 44, 9, 39]. Input frames are first passed to two feature encoders that share the same architecture to extract the correlation feature and contextual feature, respectively. Then, the multi-scale all-pairs correlation volume is calculated based on the correlation feature. Namely, given feature embeddings $\mathbf{e}_1$ and $\mathbf{e}_2$ from the target frame and the reference frame, respectively:

$$\mathbf{c}^l(i,j,m,n) = \frac{1}{2^{2l}} \sum_u^{2^l} \sum_v^{2^l} \left\langle \mathbf{e}_1(i,j), \mathbf{e}_2(2^l m + u, 2^l n + v) \right\rangle, \qquad (1)$$

where the derived $\mathbf{c}^l(i,j,m,n)$ is the average over the correlation in the local $2^l \times 2^l$ window. $l$ denotes the $l^{th}$ correlation level. $u$ and $v$ are the horizontal and vertical pixel motions, respectively. $\langle, \rangle$ refers to the dot product function. In summary, $\mathbf{c}^l(i,j,m,n)$ means the cost volume vector of $\mathbf{e}_1$ and $\mathbf{e}_2$ pooled with the $2^l \times 2^l$ kernel.

Then, the iterative decoder refines the flows via several updates. As depicted in Figure 3, flows are initialized to zeros. The derived multi-scale correlation volume, extracted context feature, and the initialized flows are passed to the decoder, and then the refinement is conducted.

## 3.2 Streamlined in-batch multi-frame pipeline

As illustrated in Figure 1, previous pairwise multi-frame networks typically perform redundant computations for video inputs, resulting in substantial computational overlap. We have briefly discussed this issue in the first two paragraphs of Section 1. Specifically, given $N$ frames $\{I_1, I_2, ..., I_N\}$ and a group size of $T$ ($N \geq T \geq 3$), pairwise methods need to form $N - T + 1$ groups, namely, $\{I_1, I_2, ..., I_T\}, \{I_2, I_3, ..., I_{T+1}\}, ..., \{I_{N-T+1}, ..., I_N\}$. In each group, pairwise methods use spatiotemporal cues from other frames to compute the optical flow only for the current frame. In this case, $T - 1$ times of forward process is performed to derive $T - 1$ flows. In contrast, StreamFlow introduces a Streamlined In-batch Multi-frame (SIM) Pipeline designed to minimize redundancy. For the same input, the SIM pipeline only forms $\lceil \frac{N-1}{T-1} \rceil$ groups, namely,

$\{I_1, I_2, ...I_T\}, \{I_T, I_{T+1}, I_{2T-1}\}, ..., \{I_{N-T+1}, ..., I_N\}$. In this case, frames are divided into non-overlapping groups (except for the first frame of each group). Within each group, the SIM pipeline calculates all uni-directional flows by modeling the spatiotemporal cues interconnecting all frames. In other words, only *one* forward pass is needed to derive $T-1$ flows. From this process, we could learn that the upper limit of the saved time for the SIM pipeline is approximately $\frac{\alpha}{T-1}$, where $\alpha$ represents the computational time of the decoder. To approach the theoretical limit as closely as possible, a lightweight yet effective method of spatiotemporal modeling is crucial. This is precisely the key contribution that StreamFlow adds to the SIM pipeline, which will be discussed in detail in Section 3.3 and Section 3.4. Additionally, when directly comparing StreamFlow with other multi-frame methods in practice, we find that the speed improvements with StreamFlow far exceed the theoretical limits calculated above. This is because StreamFlow inherently includes a memory bank mechanism in its encoder, whereas previous multi-frame methods required special additional implementation to utilize Memory banks for caching features. Nevertheless, even when compared to methods that include a memory bank, StreamFlow still achieves significant speed advantages due to its efficiency in the decoder, as demonstrated in Figure 2 and Appendix A.1.

## 3.3 Integrative spatiotemporal coherence

During the encoding process, we propose an Integrative spatiotemporal Coherence (ISC) modeling method, especially for the SIM pipeline. Our design principles for temporal modeling modules in the decoder encompass two facets: firstly, adherence to the design criteria of the SIM pipeline, with a focus on minimizing pair-wise overlap operations, such as the computation of cross-frame attention between every pair of consecutive frames. Secondly, the modules should be efficient enough and not impede the overall speed of the network.

Therefore, we design the ISC method, which introduces no additional parameters while learning spatiotemporal relations efficiently and effectively. The ISC method inherently takes the original modules in Twins. Specifically, after deriving patch embeddings from consecutive frames, patches from different frames are reorganized along the spatial dimension. Subsequently, it models the derived spatiotemporal graph using self-attention mechanisms and feed-forward layers in Twins, which can be formulated as,

$$\mathbf{x}_d^j = \mathcal{I}(\mathbf{x}_{1,d}^j, \mathbf{x}_{2,d}^j, ..., \mathbf{x}_{T,d}^j), \tag{2}$$

$$\mathbf{y}_d^j = f(\mathbf{q}(\mathbf{x}_d^j), \mathbf{k}(\mathbf{x}_d^j))\mathbf{v}(\mathbf{x}_d^j), \tag{3}$$

$$\mathbf{x}_d^j = \mathbf{x}_d^j + \mathbf{W_{proj}}\mathbf{y}_d^j, \tag{4}$$

where $f(\cdot, \cdot)$ is the attention function which conducts dot-product and softmax operation, $\mathbf{x}_{t,d}^j$ is the $j^{th}$ vector along spatial dimension at channel $d$ of the $t^{th}$ frame. $\mathcal{I}$ denotes the *integration* operation, which integrates temporally contiguous multiple input embeddings into a large feature embedding along the spatial dimension. $\mathbf{q}, \mathbf{k}$ and $\mathbf{v}$ is the derived query, key, and value vector. $\mathbf{W_{proj}}$ is the projection matrix. By leveraging the derived spatiotemporal graph, the spatial and temporal relations are learned effectively, and no additional parameters are involved.

## 3.4 Global temporal regressor

As for the decoder, we propose a Global Temporal Regressor (GTR) to predict and refine flows. Compared with the previous widely used decoder [45, 15, 44, 28, 53, 27], GTR introduces the temporal modeling module to exploit temporal cues from consecutive frames. Different from VideoFlow [38] that concatenates motion features along a temporal dimension and implicitly learns temporal relations or TransFlow [26] that applies a transformer symmetric to the encoder, the core of GTR is super convolution kernels [44] and a lightweight temporal transformer block. The input correlation volume, initialized flows, and contextual features are first passed into a motion encoder to

derive motion features and then extracted for spatiotemporal features, which can be formulated as:

$$\mathbf{m}_i^k = \mathcal{M}(\mathbf{f}_{i-1}^k, \mathbf{c}^{k,k+1}), \tag{5}$$

$$\mathbf{r}_i = \mathcal{F}_{t=1}^T(\mathbf{m}_i^t), \tag{6}$$

$$\mathbf{s}_i^k = \mathcal{A}(\mathbf{m}_i^k, \mathbf{e}^k), \tag{7}$$

$$\mathbf{g}_i^k = \mathcal{C}(\mathbf{r}_i, \mathbf{s}_i^k), \tag{8}$$

$$\mathbf{t}_i^k, \mathbf{\Delta f}_i^k = \mathcal{U}(\mathbf{m}_i^k, \mathbf{g}_i^k, \mathbf{t}_{i-1}^k), \tag{9}$$

$$\mathbf{f}_i^k = \mathbf{f}_{i-1}^k + \mathbf{\Delta f}_i^k \tag{10}$$

where $\mathbf{m}_i^k$ is the derived motion feature of frame $k$ at the $i^{th}$ update and $\mathbf{f}_{i-1}^k$ denotes the flow of frame $k$ after $i-1^{th}$ refinement. $\mathbf{c}^{k,k+1}$ denotes the correlation volume between frame $k$ and $k+1$. $\mathcal{M}$ denotes the motion encoder which is the same as that in SKFlow [44]. $\mathbf{r}_i$ denotes the temporal feature embedding extracted from the motion features of all frames. Notably, the caching mechanism of the MemoryBank is employed, thus necessitating the calculation of $\mathbf{r}_i$ only once for different frames. $\mathcal{F}$ is a lightweight temporal-learning layer that consists of temporal attention and feed-forward layers. $\mathbf{e}^k$ refers to the feature embedding of frame $k$. Note that $e^k$ and $c^{k,k+1}$ are not updated during the refinement. $\mathcal{A}$ denotes the spatial cross attention inspired by [15], but takes $\mathbf{m}_i^k$ and $\mathbf{e}^k$ as the input. $\mathcal{C}$ represents the concatenation operation and $\mathcal{U}$ refers to the motion updater. $\mathbf{t}_i^k$ denotes the extracted contextual information, which will be updated during each refinement. In practice, the decoder estimates the residual of flow $\mathbf{\Delta f}_i^k$. And the final flow $\mathbf{f}^k$ is updated via $\mathbf{\Delta f}_i^k$ during each refinement.

### 3.5 Supervision

StreamFlow adopts the overall loss in the same group as the total loss function. For each flow, StreamFlow adopts the same loss function as successful two-frame networks. Namely, the weighted sum for the predicted flows at different refinements. During both the training and the fine-tuning process, the supervision can be formulated as follows:

$$\mathcal{L} = \sum_{k=1}^{T-1} \sum_{i=1}^{N} \theta^{N-i} \, ||\mathbf{f}_i^k - \mathbf{f}_{gt}^k||_1, \tag{11}$$

where $\mathbf{f}_i^k$ refers to the flow of frame $k$ at the $i^{th}$ refinement. $T$ and $N$ are the number of frames and refinements, respectively. $\theta$ denotes the weights on corresponding estimated flows. $\mathbf{f}_{gt}$ is the ground truth flow and $|| \cdot ||_1$ means the $l_1$ distance between ground truth and our predicted flow. In practice, $N$ is set to 12, $\theta$ is set to 0.8, the same as previous works [38, 45, 44, 15] for a fair comparison.

## 4 Experiments

**Experimental setup.** In this study, we evaluate StreamFlow on the Sintel [3], KITTI [31], and Spring [30] datasets, following previous works [44, 9, 7]. In previous works, models are initially pre-trained on the FlyingChairs [8] and FlyingThings [29] datasets using the "C+T" schedule and then are subsequently fine-tuned using the "C+T+S+K+H" schedule on Sintel and KITTI datasets. *After "C+T", the cross-dataset generalization tests are often performed, and the models from this stage are frequently used for various cross-domain video downstream tasks [18, 23]. In "+S+K+H", the evaluation typically focuses on intra-domain generalization capabilities.* In specific, for Sintel, models are trained on a combination of FlyingThings, Sintel, KITTI, and HD1K [17]. Models are then trained on the KITTI dataset for KITTI evaluation and on the Spring dataset for Spring evaluation.

**Implementation details.** Our StreamFlow method is built with PyTorch [34] library, and our experiments are conducted on the NVIDIA A100 GPUs. During training, we adopt the AdamW [24] optimizer and the one-cycle learning rate policy [40], following previous works [45, 15, 44]. The number of refinements in the decoder is set to 12, following previous works. Given the absence of multi-frame data information in the Chairs dataset, we follow VideoFlow [38] to directly train on the FlyingThings in the first stage. For the Spring dataset, we follow the settings of MemFlow [7] and fine-tune the model for 180k steps. The remaining training configurations are consistent with prior works [38, 44, 15, 45]. The temporal and non-temporal modeling modules are concurrently trained.

| Training Data | Method | Sintel (train) | | KITTI-15 (train) | | Sintel (test) | | KITTI-15 (test) |
|---|---|---|---|---|---|---|---|---|
| | | Clean | Final | Fl-epe | Fl-all | Clean | Final | Fl-all |
| (C)+T | HD3 [52] | 3.84 | 8.77 | 13.17 | 24.0 | - | - | - |
| | PWC-Net [42] | 2.55 | 3.93 | 10.35 | 33.7 | - | - | - |
| | RAFT [45] | 1.43 | 2.71 | 5.04 | 17.4 | - | - | - |
| | CRAFT [41] | 1.27 | 2.79 | 4.88 | 17.5 | - | - | - |
| | AGFlow [28] | 1.31 | 2.69 | 4.82 | 17.0 | - | - | - |
| | Separable Flow [53] | 1.30 | 2.59 | 4.60 | 15.9 | - | - | - |
| | GMA [15] | 1.30 | 2.74 | 4.69 | 17.1 | - | - | - |
| | SKFlow [44] | 1.22 | 2.46 | 4.27 | 15.5 | - | - | - |
| | FlowFormer [9] | 1.00 | 2.45 | 4.09 | 14.7 | - | - | - |
| | GAFlow [27] | 1.02 | 2.45 | 3.98 | 15.0 | - | - | - |
| | TransFlow [26] | 0.93 | 2.33 | 3.98 | 14.4 | - | - | - |
| | VideoFlow-BOF [38] | 1.03 | 2.19 | 3.96 | 15.3 | - | - | - |
| | SplatFlow [46] | 1.22 | 2.97 | **3.70** | 15.3 | - | - | - |
| | **Ours** | **0.87** | **2.11** | 3.85 | **12.6** | - | - | - |
| (C)+T+S+K+H | IRR-PWC [12] | (1.92) | (2.51) | (1.63) | (5.3) | 3.84 | 4.58 | 7.65 |
| | MaskFlowNet [54] | - | - | - | - | 2.52 | 4.17 | 6.10 |
| | Separable Flow[53] | (0.69) | (1.10) | (0.69) | (1.6) | 1.50 | 2.67 | 4.64 |
| | PWC-Fusion [43] | - | - | - | - | 3.43 | 4.57 | 7.17 |
| | StarFlow [1] | - | - | - | - | 2.72 | 3.71 | 7.65 |
| | RAFT⋆ [45] | (0.76) | (1.22) | (0.63) | (1.5) | 1.61 | 2.86 | 5.10 |
| | GMA⋆ [15] | (0.62) | (1.06) | (0.57) | (1.2) | 1.39 | 2.47 | 5.15 |
| | GMFlowNet [55] | (0.59) | (0.91) | (0.64) | (1.5) | 1.39 | 2.65 | 4.79 |
| | AGFlow⋆ [28] | (0.65) | (1.07) | (0.58) | (1.2) | 1.43 | 2.47 | 4.89 |
| | SKFlow⋆ [44] | (0.52) | (0.78) | (0.51) | (0.9) | 1.28 | 2.27 | 4.84 |
| | FlowFormer [9] | (0.48) | (0.74) | (0.53) | (1.1) | 1.16 | 2.09 | 4.68 |
| | MFRFlow [16] | (0.64) | (1.04) | (0.54) | (1.1) | 1.55 | 2.80 | 5.03 |
| | MFCFlow [4] | (0.56) | (0.89) | (0.55) | (1.1) | 1.49 | 2.58 | 5.00 |
| | TransFlow [26] | (0.42) | (0.69) | (0.49) | (1.05) | 1.06 | 2.08 | 4.32 |
| | VideoFlow-BOF [38] | (0.37) | (0.54) | (0.52) | (0.85) | **1.00** | **1.71** | 4.44 |
| | SplatFlow [46] | (0.53) | (0.91) | (0.80) | (2.40) | 1.12 | 2.07 | 4.61 |
| | **Ours** | **(0.28)** | **(0.38)** | **(0.47)** | **(0.77)** | 1.04 | 1.87 | **4.24** |

Table 1: Quantitative results on Sintel and KITTI. The average End-Point Error (EPE) is reported as the evaluation metric if not specified. ⋆ refers to the warm-start strategy [45] that use the previous flow for initialization. Bold and underlined metrics denote the method that ranks 1st and 2nd, respectively.

## 4.1 Quantitative results

From Table 1 and Table 4.1, we can learn that StreamFlow achieves advanced 0-shot performance on Sintel and KITTI. Compared to previous methods, StreamFlow reduces the 0-shot end-point error by 0.16 and 0.08 on the challenging Sintel clean and final pass, respectively. On KITTI, StreamFlow outperforms the previous state-of-the-art 0-shot results with 0.11 and 17.65% lower EPE and Fl-all metric. Besides, without self-supervised pre-training or bi-directional flows, StreamFlow attains commendable accuracy and efficiency on the challenging Sintel and KITTI test benchmarks using (C)+T+S+K+H schedule. On the challenging Spring [30] dataset, StreamFlow also achieves enhanced performance both before and after fine-tuning. **A detailed analysis on its shortcomings is in Section 5**.

## 4.2 Occlusion analysis

In this section, we validate if StreamFlow could help improve the performance on the occlusions. We compare StreamFlow with its base two-frame model Twins-SKFlow, which strengthens SKFlow [44] with the Twins [5] encoder. Evaluations are conducted on the matched and unmatched areas of the Sintel test dataset. The matched areas denote regions visible in adjacent frames and the unmatched areas refer to regions visible only in one of two adjacent frames. Our models are trained using the T+S+H+K schedule. We could learn that StreamFlow attains remarkable improvements on occluded areas, as shown in Table 3. We also visualize the performance on occluded regions, which are shown in the supplements. On the challenging Sintel final test set, StreamFlow attains the improvement of 10.77% and 11.83% on unmatched and matched regions, respectively. On the clean pass, StreamFlow improves the performance by 15.53%, 15.56%, and 15.45% on unmatched, matched, and overall

| Method | 1px | | | | | | | | | | | | EPE | Fl | WAUC |
|---|---|---|---|---|---|---|---|---|---|---|---|---|---|---|---|
| | total | low-det. | high-det. | matched | unmat. | rigid | non-rig. | not-sky | sky | s0-10 | s10-40 | s40+ | | | |
| RAFT[†] [45] | 6.79 | 6.43 | 64.09 | 6.00 | 39.48 | 4.11 | 27.09 | 5.25 | 30.18 | 3.13 | 5.30 | 41.40 | 1.48 | 3.20 | 90.92 |
| GMA[†] [15] | 7.07 | 6.70 | 66.20 | 6.28 | 39.89 | 4.28 | 28.25 | 5.61 | 29.26 | 3.65 | 5.39 | 40.33 | 0.91 | 3.08 | 90.72 |
| GMFlow[†] [49] | 10.36 | 9.93 | 76.61 | 9.06 | 63.95 | 6.80 | 37.26 | 8.95 | 31.68 | 5.41 | 9.90 | 52.94 | 0.95 | 2.95 | 82.34 |
| FlowFormer[†] [9] | 6.51 | 6.14 | 64.22 | 5.77 | 37.29 | 3.53 | 29.08 | 5.50 | 21.86 | 3.38 | 5.53 | 35.34 | 0.72 | 2.38 | 91.68 |
| MS-RAFT+[†] [14] | 5.72 | 5.37 | 61.50 | 5.04 | 33.95 | 3.05 | 25.97 | 4.84 | 19.15 | 2.06 | 5.02 | 38.32 | 0.64 | 2.19 | 92.89 |
| MemFlow[†] [7] | 5.76 | 5.39 | 63.35 | 5.11 | 32.76 | 3.29 | 24.42 | 4.49 | 24.99 | 2.92 | 4.82 | 32.07 | 0.63 | 2.11 | 92.25 |
| Ours[†] | **5.22** | **4.87** | **59.55** | **4.56** | **32.34** | **2.87** | **23.00** | **4.44** | 17.06 | **2.60** | **4.49** | 29.07 | **0.61** | **1.86** | **93.25** |
| CroCo-Flow [48] | 4.57 | 4.21 | **60.59** | 3.85 | **34.20** | 2.19 | 22.50 | 4.48 | **5.87** | 1.23 | 4.33 | 33.13 | 0.50 | 1.51 | 93.66 |
| MemFlow [7] | 4.48 | 4.12 | 61.70 | 3.74 | 35.12 | 2.39 | **20.31** | **3.93** | 12.81 | 1.31 | 4.44 | 31.18 | **0.47** | **1.42** | 93.86 |
| Ours | **4.15** | **3.79** | 61.30 | **3.42** | 34.30 | **1.99** | 20.54 | 3.99 | 6.68 | 1.24 | 4.38 | **27.94** | **0.47** | **1.42** | **94.40** |

Table 2: Quantitative results on Spring test benchmark. Measures are from the official Spring website, including the total score, EPE, Fl, WAUC, and detailed metrics such as 1px outlier rate, etc. Important metrics are highlighted. [†] denotes 0-shot test using the checkpoint from "C+T+S+H+K".

| Method | Clean | | | Final | | |
|---|---|---|---|---|---|---|
| | Unm. | Mat. | All | Unm. | Mat. | All |
| GMFlow [49] | 10.56 | 0.65 | 1.74 | 15.80 | 1.32 | 2.90 |
| GMFlowNet [55] | 8.49 | 0.52 | 1.39 | 13.88 | 1.27 | 2.65 |
| SKFlow [44] | 7.24 | 0.55 | 1.28 | 11.51 | 1.46 | 2.28 |
| FlowFormer [9] | 7.16 | 0.42 | 1.16 | 11.30 | 0.96 | 2.09 |
| TransFlow [26] | 6.77 | **0.36** | 1.06 | 10.96 | 0.99 | 2.08 |
| Baseline | 7.60 | 0.45 | 1.23 | 11.70 | 0.93 | 2.11 |
| Ours | **6.42** | 0.38 | **1.04** | **10.44** | **0.82** | **1.87** |

Table 3: Occlusion analysis on Sintel test set. Unm. and Mat. denote performance on unmatched and matched areas, respectively. "Baseline" denotes our baseline method Twins-SKFlow.

regions, respectively. From the table, we can learn that StreamFlow improves not only the flow estimation in unmatched regions but also the estimation in matched regions.

## 4.3 Ablations

In this section, we verify the effectiveness of StreamFlow designs, as shown in Table 4. For a fair comparison, all models are trained with the same settings on the FlyingThings dataset. Then we evaluate each method on Sintel and KITTI. Below we will introduce each experiment in detail.

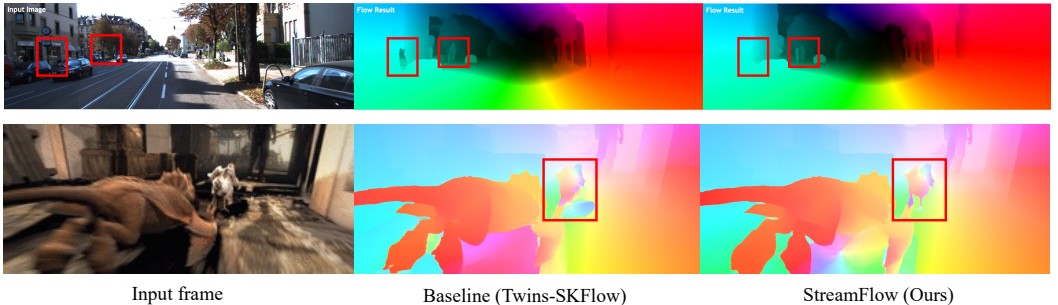

| Input frame | Baseline (Twins-SKFlow) | StreamFlow (Ours) |

Figure 4: Visualizations of results on Sintel and KITTI test sets. Differences are highlighted with red bounding boxes. StreamFlow achieves fewer artifacts on both synthetic and real-world scenes. More visualization results on DAVIS [35] and occluded regions are in the *supplements*.

**SIM pipeline.** We test the efficiency of the vanilla pairwise pipeline and our SIM pipeline. Pairwise methods utilize multi-frames to predict the flow of the current two frames and bring substantial redundant computation, while the SIM pipeline estimates multiple flows concurrently and minimizes the overlapping calculation. As shown in Table 4, the SIM pipeline brings great gain in efficiency. Notably, there might be information loss for the SIM pipeline due to different frame distances. For instance, given the frames $I_{t-1}, I_t, I_{t+1}$, the estimated flow $F_{t-1 \to t}$ is typically less accurate than that derived from the sequence $I_{t-2}, I_{t-1}, I_t$. However, this issue tends to be alleviated with longer sequences. When the number of frames increases to 4, the impact on accuracy is significantly reduced. We will discuss this issue in more detail in the appendix.

**Temporal modules.** In this part, we explore the performance and efficiency of different temporal modeling methods in the flow encoder. Temporal attn refers to applying a temporal attention layer after each spatial self-attention modeling in Twins. Pseudo conv [36] denotes stacking 1D convolution layers in the temporal dimension to imitate 3D convolutions at minimal cost. We also apply 3D convolutions at the end of the flow encoder to learn temporal relations. As shown in Table 4, our ISC module achieves a good trade-off between efficiency and effectiveness. The improvements achieved by other methods are not as pronounced. We hypothesize that the limited volume of optical flow data impedes the efficient training of the spatiotemporal module from scratch to accomplish good optimization. For comparison, VideoFlow does not apply temporal modeling modules in the encoder, and TransFlow [26] applies self-supervised pre-training for better optimization.

**Extra parameters.** In this part, we aim to determine whether the performance gain is due to the additional parameters or the effective temporal modeling method. To this end, we introduce the additional parameters by widening the baseline network. Namely, we extract higher-dimension features along the spatial dimension and concatenate them with the original motion feature. All models in this section are equipped with the ISC module. "w/o" denotes the baseline Twins-SKFlow network. "w" means adding additional parameters. "Ours" denotes the method equipped with our temporal modeling modules. Results show the improvement achieved by simply adding more parameters is minor, and the performance gain is primarily attributed to the effectiveness of StreamFlow modules.

| Experiment | Method | Sintel | | | | KITTI | | Param | Latency |
|---|---|---|---|---|---|---|---|---|---|
| | | Clean | Final | Occ | Noc | Fl-epe | Fl-all | (M) | (ms) |
| SIM pipeline | w/o | 1.03 | 2.34 | 7.69 | 0.35 | 4.64 | 14.70 | 12.49 | 122.18 |
| | w/ | 1.03 | 2.34 | 7.69 | 0.35 | 4.64 | 14.70 | 12.49 | **84.59** |
| Tem. modules | w/o | 1.03 | 2.34 | 7.69 | 0.35 | 4.64 | 14.70 | **12.49** | **84.59** |
| | Tem. attn | **0.96** | 2.31 | 7.38 | 0.35 | 4.38 | 14.96 | 14.14 | 91.17 |
| | Pse. 3D conv | 1.05 | 2.36 | 7.60 | 0.38 | 4.46 | 15.20 | 13.48 | 87.41 |
| | 3D conv | 0.98 | 2.34 | 7.63 | 0.33 | 4.57 | 15.59 | 16.03 | 93.05 |
| | ISC | 0.97 | **2.29** | **7.11** | **0.32** | **4.14** | **14.16** | 12.49 | 88.35 |
| Extra params | w/o | 0.97 | 2.29 | 7.11 | 0.32 | 4.14 | 14.16 | **12.49** | **84.59** |
| | w/ | 0.98 | 2.24 | 7.33 | **0.31** | 4.15 | 13.94 | 13.77 | 89.29 |
| | Ours | **0.93** | **2.15** | **7.06** | **0.31** | **3.92** | **12.36** | 13.77 | 89.76 |
| GTR module | w/o | 0.97 | 2.29 | 7.11 | 0.32 | 4.14 | 14.16 | **12.49** | **88.35** |
| | w/ | 0.93 | **2.15** | **7.06** | **0.31** | **3.92** | **12.36** | 13.77 | 89.76 |
| ISC module | w/o | 1.01 | 2.19 | 7.23 | 0.33 | 4.06 | 13.95 | 13.77 | **86.02** |
| | w/ | **0.93** | **2.15** | **7.06** | **0.31** | **3.92** | **12.36** | 13.77 | 89.76 |
| # Frames | 3 | 0.93 | 2.15 | 7.06 | 0.31 | 3.92 | **12.36** | **13.77** | 89.76 |
| | 4 | **0.87** | **2.11** | **6.24** | 0.31 | **3.85** | 12.62 | 14.25 | **85.53** |

Table 4: Ablations on our proposed design. All models are trained using the "C+T" schedule. The number of refinements is 12 for all methods. The settings used in our final model are underlined.

**GTR module.** We also examined whether the GTR module could enhance flow predictions. "w/o" means applying vanilla SKFlow decoder while "w" denotes using GTR. All models in this part utilize the ISC module in the encoder. Table 4 demonstrates the necessity of incorporating the GTR, which could achieve stable improvement on multiple benchmarks. We could learn that GTR especially helps

the estimation on the challenging final pass, with the performance gain of $0.14$. In *supplements*, we give a more detailed discussion of its initialization, which is key to its training process.

**ISC module.** We verify the effectiveness of the proposed ISC module. All models in this part adopt GTR as the flow decoder. From Table 4, we could learn that the ISC module is efficient and effective in temporal modeling and greatly contributes to the improvement of the pipeline. It introduces no additional parameters and a modest increase in runtime, while significantly boosting the performance.

**# Frames.** We delve into the influence of different numbers of frames, as illustrated in Table 4. We set the number of frames to 4 due to limitations in GPU memory. From an efficiency standpoint, augmenting the number of frames results in a higher proportion of redundant computations eliminated by StreamFlow, consequently leading to a more substantial improvement in processing time. Although there is an increase in the parameter count for temporal modeling, the efficiency is further enhanced in the context of four input frames due to a reduced proportion of redundant computations, resulting in a shorter average prediction time per frame compared to the three-frame setting.

## 4.4 Qualitative results

We demonstrate visualization results on both synthetic (Sintel [3]) and real-world scenes (KITTI [31]), as shown in Figure 4. In the *supplements*, we also show the visualizations on the real-world dataset DAVIS [35]. Our models are pre-trained using the T+H+S+K schedule. We could learn that StreamFlow could still achieve remarkable qualitative results when generalized to real-world scenes.

## 4.5 Efficiency analysis

We evaluate the efficiency of the StreamFlow in terms of runtime and parameter counts. Our experiments were conducted on NVIDIA A100 GPUs. Models are trained using the (C+)T schedule and evaluated on the Sintel dataset. The runtime is measured as the average inference time per frame of five runs on the Sintel training set. We could learn StreamFlow achieves comparable efficiency with state-of-the-art two-frame methods while achieving superior performance. The key to maintaining high efficiency is its SIM pipeline. StreamFlow does not perform pairwise redundant computation and predicts all flows simultaneously. Another reason for the high speed is its CNN-based decoder. We could learn that StreamFlow is much faster than the pure two-frame transformer architecture FlowFormer. Besides, the specially designed lightweight temporal modules also contribute to the performance, simultaneously aiding in better results compared to the 2-frame baseline Twins-SKFlow.

# 5    Limitations

StreamFlow faces two primary challenges: (1) GPU memory usage in training. Although it is not an issue during inference (e.g., with a $432 \times 1024$ input, it needs about only 2.4 G and 3.3 G when the # frame is 3 and 4, respectively). But it is significantly increased during training. The storage of gradients and the batch size cause the GPU memory on a single card to reach approximately 40 G in a 4-frame setting. (2) Inter-group cues utilization. Limited to the SIM pipeline, StreamFlow is confined to using only intra-group information, and it does not utilize inter-group information during modeling. Despite achieving commendable results and surpassing some methods that incorporate inter-group information, bidirectional flows, or self-supervised pre-training strategies in cross-dataset tests, how to address this while maintaining good efficiency is a worthwhile issue to consider in future work.

# 6    Conclusion

In this work, we proposed StreamFlow, a multi-frame optical flow estimation approach proficient in estimating optical flows across multiple video frames using efficient spatiotemporal relationship mining. StreamFlow aims to estimate consecutive unidirectional optical flows with less overlapping computation. It proposes to estimate multi-frame optical flows via the proposed SIM pipeline and introduces efficient and effective ISC and GTR methods for temporal modeling under such circumstances. Extensive experiments on multiple challenging benchmarks demonstrate the efficiency and effectiveness of the proposed StreamFlow method.

**Acknowledgements.** This work was supported by The Major Key Project of PCL (PCL2024A02), Natural Science Foundation of China (62271013, 62031013), Guangdong Province Pearl River Talent Program (2021QN020708), Guangdong Basic and Applied Basic Research Foundation (2024A1515010155), Shenzhen Science and Technology Program (JCYJ20230807120808017), Shenzhen Fundamental Research Program (GXWD20201231165807007-20200806163656003), and Sponsored by CAAI-MindSpore Open Fund, developed on OpenI Community (CAAIXSJLJJ-2023-MindSpore07).

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

# A  Appendix

## A.1  Comparison of latency using memory bank

As discussed in Section 3.2, to better demonstrate that the temporal advantage of StreamFlow is not solely due to the memory bank, this section explores the efficiency comparison between StreamFlow and other methods when using a memory bank. Given that the model's runtime is closely related to the coding implementation, this comparison prioritizes officially open-sourced multi-frame optical flow methods. However, as of the writing of this paper, the choices for leading open-source multi-frame methods are quite limited, and thus VideoFlow [38] was selected for comparison. The experimental setup and the machine are consistent with those described in Section 4.5, and the measured time is the average of five tests. The input is resized to $432 \times 1024$, and the model is trained via (C+)T manner. As shown in Table A.1, it can be observed that StreamFlow still exhibits good efficiency in time. This is because, in addition to the memory bank, it further optimizes the average estimation time in the decoder.

| Method | Sintel (clean) | Sintel (final) | Fl-EPE | Fl-all | Latency | Hardware |
|---|---|---|---|---|---|---|
| VideoFlow-BOF [38] | 1.03 | 2.19 | 3.96 | 15.3 | 122.37ms | A100-40G |
| StreamFlow (Ours) | **0.87** | **2.11** | **3.85** | **12.6** | **85.53ms** | A100-40G |

Table 5: Comparison of latency using memory bank.

## A.2  Impact of the frame distance

Within the SIM pipeline, the impact of different frame distance $\mathbf{D}$ indeed exists. For instance, the flow $F_{t-1,t}$ derived from $I_{t-2}, I_{t-1}, I_t$ tends to be more accurate than that from $I_{t-1}, I_t, I_{t+1}$ due to the different distance $\mathbf{D}_{t-1,t-2}$ and $\mathbf{D}_{t-1,t+1}$. As the frame distance increases, the information provided may decrease, as confirmed by the results in Table 7. However, this impact might not necessarily grow larger with more frames, which could be due to the affected frames mainly being distributed at the head or the tail of a group. We could define the longest frame distance that provides effective information as $m$. As the length of the group increases, there will be more frames in the middle of the group (i.e., frames $I_t$ within the interval $[I_{t-m}, I_{t+m}]$ all lying within the group), and fewer frames distributed on both ends. As shown in Table 7, the impact was weakened with 4 frames. Maybe the future study on an appropriate choice of $m$ is helpful for future multi-frame optical flow work.

## A.3  Detailed metrics on the Sintel test set

Table A.4 presents a more detailed comparison on the Sintel test set between StreamFlow and its two-frame baseline. The detailed metrics including all/matched/unmatched EPE, d0-10, d10-60, d60-140, s0-10, s10-40, s40+ are from the official Sintel website. We could learn that StreamFlow performs exceptionally well in unmatched areas, validating its effectiveness in addressing occlusion issues. Compared to the baseline method, it improved by 15.53% and 10.77% in the Matched areas on the clean pass and final pass, respectively, and by 15.56% and 11.83% in the Unmatched areas. Within different ranges of occlusion boundaries, it showed improvements of 15.13% and 10.30% at d0-10, 15.79% and 13.51% at d10-60, and 10.00% and 12.50% at d60-140 on the clean pass and final pass, respectively. For pixels moving at different speeds, it improved by 16.00% and 14.58% at s0-10, 10.39% and 10.14% at s10-40, and 14.41% and 10.49% at s40+ on the clean pass and final pass, respectively. It is shown that StreamFlow has outperformed its baseline on both the clean pass and final pass, with a more pronounced improvement on the clean pass. The enhancements are particularly noticeable in areas of occlusion and for objects moving at high speeds.

## A.4  Relationship between image size and GPU memory

In this section, we show the GPU usage of StreamFlow. During the inference, without the optimization in the third-party package, StreamFlow takes about 2.4 G and 3.3 G for $432 \times 1024$ inputs with the number of frame set to 3 and 4, respectively. With PyTorch 2.2 and flash-attention, using 12

refinements and 4 frames, the GPU memory usage for StreamFlow is shown in Table **??**. Specifically, when the image size increases by 4 times, the GPU memory usage increases to nearly 4.4 times. When the image size is increased by 9 times, GPU memory usage grows to nearly 20 times. For most current scenarios, StreamFlow still maintains a relatively moderate use of GPU memory. The variation in GPU memory usage may be influenced by underlying optimizations in the framework. We believe the memory usage could be further optimized in the future.

## A.5  Qualitative analysis on real-world scenes

In this section, we facilitate our visualizations and evaluations using two prominent real-world datasets, namely DAVIS [35]. The DAVIS dataset, short for Densely Annotated VIdeo Segmentation, is a widely recognized benchmark in the field of computer vision. It comprises high-quality video sequences captured in diverse scenarios, encompassing a broad range of challenging visual conditions such as occlusions, motion blur, and dynamic object interactions. The dataset provides pixel-level annotations for every frame, facilitating precise evaluation and comparison of various video segmentation methods. The visualizations on the DAVIS dataset are shown in Figure 5. Our model is pretrained using the "T" and "T+S+H+K" schedule and then fine-tuned on KITTI [31]. "T" denotes the FlyingThings [29] dataset and "T+S+H+K" refers to the combination of the FlyingThings, Sintel [3], HD1K [17], and KITTI datasets. Then we infer our models on the DAVIS dataset. The number of refinements is set to 12. The number of input frames for each non-overlapping group is 3. We could learn that StreamFlow demonstrates remarkable adaptability across real-world datasets, showing its robust performance in challenging scenes for optical flow estimation. This is particularly evident in scenarios such as the occlusion of the bear's hind legs in the first row, first column, and the small motion of the small tennis ball in the last column. Additionally, it can be observed that in the motion captured in the first row, second, and third columns, the hind legs of the camel and the leg movements of the dancer are also vividly delineated. These instances reaffirm its efficacy in diverse and demanding environments for optical flow estimation.

## A.6  Qualitative analysis on occluded regions

In this section, we focus on the performance of the occluded regions. As discussed in previous works [15, 44], here we term occlusions as areas where pixels appear in the current frame while disappearing in the next frame. We visualize the flow-error map on occluded regions of the Sintel dataset with the official occlusion masks. All models are trained using the (C+)T schedule. As shown in Figure 6, significant occluded areas are highlighted using red boxes. A darker color in the flow-error map denotes a more significant error. We could learn that StreamFlow achieves better overall performance, and attains leading performance on the occluded regions.

| Image Size | GPU Memory |
|---|---|
| $360 \times 640$ | 1.19 G |
| $720 \times 1280$ | 5.20 G |
| $1080 \times 1920$ | 24.11 G |

Table 6: Relationship between input size and GPU memory usage during inference. The number of frames is set to 4.

| Method | Sintel | |
|---|---|---|
| | clean | final |
| T3 | 0.93 | 2.15 |
| T3[†] | 0.90 | 2.13 |
| T4 | 0.87 | **2.11** |
| T4[†] | **0.86** | **2.11** |

Table 7: Impact of frame distance. [†] denotes using nearer frames.

| Data | Method | Unm. | Mat. | All | d0-10 | d10-60 | d60-140 | s0-10 | s10-40 | s40+ |
|------|--------|------|------|-----|-------|--------|---------|-------|--------|------|
| Clean | Baseline | 7.60 | 0.45 | 1.23 | 1.19 | 0.38 | 0.20 | 0.25 | 0.77 | 7.01 |
|  | Ours | **6.42** | **0.38** | **1.04** | **1.01** | **0.32** | **0.18** | **0.21** | **0.69** | **6.00** |
| Final | Baseline | 11.70 | 0.93 | 2.11 | 2.33 | 0.74 | 0.40 | 0.48 | 1.38 | 11.92 |
|  | Ours | **10.44** | **0.82** | **1.87** | **2.09** | **0.64** | **0.35** | **0.41** | **1.24** | **10.67** |

Table 8: Results on Sintel test set. Unm. and Mat. denote performance on unmatched and matched areas, respectively. "Baseline" denotes our baseline method Twins-SKFlow.

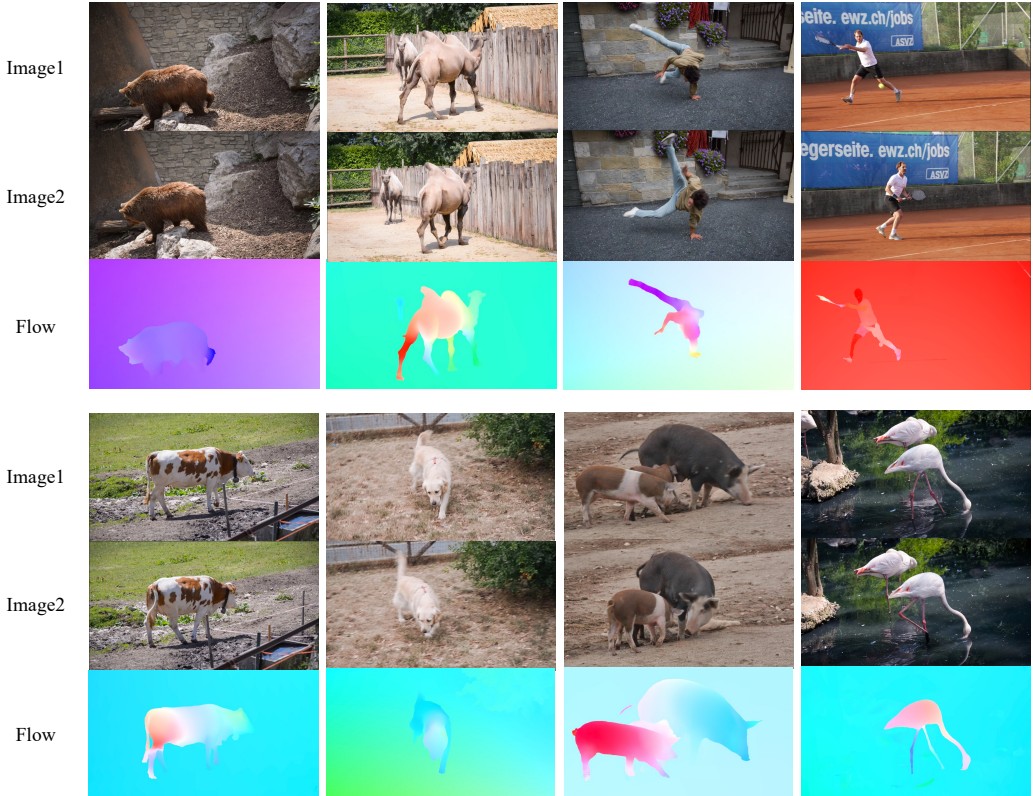

Figure 5: Visualizations of predicted flows on DAVIS [35]. StreamFlow demonstrates robust generalization to other real-world datasets, performing well in challenging scenarios for optical flow estimation, as evidenced by instances such as the occluded hind legs of the bear in the first column and the small tennis ball in the last column.

## A.7 Initialization of GTR

In this section, we investigate the impact of different GTR initialization methods. Previous works in spatio-temporal modeling such as [2] have suggested initializing the temporal modules with zero values. We employed two distinct initialization approaches, namely zero initialization and PyTorch's default initialization, and the corresponding results are presented in Table 9. Following training on the FlyingThings dataset, the model was tested on the Sintel and KITTI datasets. It is evident from the results that zero initialization could contribute to a better overall performance.

| Method | Sintel (Clean) | Sintel (Final) | KITTI (EPE) | KITTI (Fl-all) |
|--------|----------------|----------------|-------------|----------------|
| Default | **0.91** | 2.20 | 4.05 | 13.44 |
| Zero-init | 0.93 | **2.15** | **3.92** | **12.36** |

Table 9: Comparison of different ways of initialization. All models are trained under the FlyingThings.

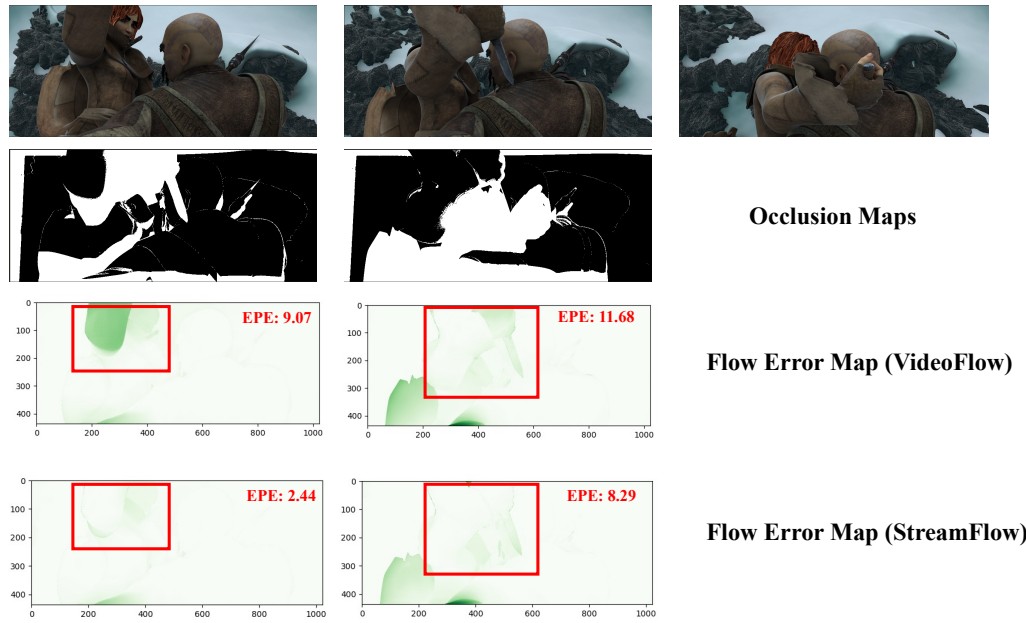

Figure 6: Visualizations of the performance on the occluded regions. StreamFlow achieves comparable performance even with advanced methods. All models are trained on the FlyingThings dataset. A darker color in the flow error map denotes a higher estimation error compared with ground truth.

