# OpenReview forum: "StreamFlow: Streamlined Multi-Frame Optical Flow Estimation for Video Sequences"
_NeurIPS.cc/2024/Conference — NeurIPS 2024 poster_

### Official Review · Reviewer_ba6H · 2024-07-08

**Soundness:** 4
**Presentation:** 3
**Contribution:** 3
**Rating:** 6
**Confidence:** 5

**Summary:**

The paper proposes a multi-frame optical flow estimation model with a novel flow decoder that estimates a flow output for all frames simultaneously. The paper describes this process as  a Stream-lined In-batch Multiframe (SIM) pipeline and argues that this leads to efficiency gains when processing video as each multi-frame batch needs to contain each video frame only once, i.e., the video processing can advance by multi-frame batch instead of frame-by-frame. The decoder is termed Global Temporal Regressor and uses an iterative design as common in many recent optical flow methods. In order to estimate multi-flow output the decoder uses a correlation volume for each pair of input images. The first step of the decoder is the output of a "MotionEncoder" which uses the current flow estimate and the correlation volume. These motion features form the input to a temporal feature encoder which is implemented with the Super Convolution Kernels by Sun et al., 2022. Spatial image features are than matched with a cross attention module and also input in the updater for the flow. The paper explains the stacking of feature over time and using the twins transformer for encoding temporal relationship in their Integrative Spatio-temporal Coherence (ISC) modeling. The paper reports experiments with the KITTI and MPI-Sintel datasets and two training schedules. The tables show that the proposed StreamFlow achieves either SOTA or close to SOTA performance. The paper reports that A100 GPU with 40GB of memory is enough to train either the method to work with 3 or 4 images as input to the batch. The paper reports a more detailed breakdown of Sintel results and an ablation study is provided.

**Strengths:**

The stream processing for optical flow is new to the best of my knowledge. In the proposed computationally heavy method, it enables the distribution of the computational effort over multiple output. This idea brings the method to a computational level per frame which is similar to methods such as RAFT. The idea of video processing in batches with multiple outputs has been used in MTTR, a referring video object segmentation by Botach et al., 2022.

The proposed flow decoder GTR is novel as it works with multiple correlation volumes, uses spatio-temporal features as inputs and combines them to an iterative design.

The paper also proposes to append temporal input in space for spatio-temporal feature encoding in the ISC that are the "appearance features" for the flow decoder.

The results shows that the proposed StreamFlow is very capable in estimating flow in the Sintel and KITTI datasets with a training schedule of only Chairs and Things.

The paper contains an ablation study to evaluate the components of the proposed model.

**Weaknesses:**

The multiple input - multiple output processing does only partially address the high computation cost of recent optical flow methods. In particular the high memory costs limits the use of StreamFlow to non-edge devices. Considering that optical flow is a low-level visual cue, this prevents StreamFlow to be used as an optical flow sub-module in other tasks.

The paper overstates the performance gains of the method. In particular, considering the full training schedule StreamFlow does not reach top performance on MPI-Sintel and in Kitti. While the table shows top performance on Kitti, the table does not include the on-line videoflow MOF method which achieves an F1-all score of 4.08 (Section 3 of VideoFlow supplemental material).

The paper does not conduct any experiments with data other than MPI-Sintel and Kitti. The recent large image Spring dataset by Mehl et al., 2023 would be a good choice.

The authors did not include consideration of MemFlow, CVPR 2024 by Dong and Fu which has been available on arXiv since April 2024. As one may argue that this is parallel work, I think their superior results raise questions about the design choices for the proposed method.

Unrolling time into space in the ISC module seems to be a fairly standard trick. It is hard to think of it as a novel contribution, even though this may be the first time that it has been used in optical flow.

**Questions:**

Is there a relationship between image size and memory requirements and if so can the method be used to process large-scale images, e.g., in Spring?

The MPI-Sintel datasets makes numerous measures for detailed analysis of the flow results available. Can these measures be listed and discussed in the supplemental material?

The efficiency analysis makes claims about the efficiency of StreamFlow but no such data is provided. The only information included seems the number of parameters and the latency of the method. No comparison to other methods is given except with VideoFlow-BOF on latency in the Table 4 (Appendix A.1).

The training EPE is very low. Does this indicate overfitting?

The paper needs some careful proof-reading:
l.27 The consecutive flow goes backwards in time. Please check notation.
l.48 "the aim of bidirectional flows" cannot be understood.
l.83 "features from different in multiple scales" missing word.
l.122 "iterative decoder is the paradigm proposed in RAFT" has been proposed earlier by Hur and Roth in their IRR-PWC, 2019 paper.
l.177,184,200 "could" -> can
Eqn. 2 is very confusing by typing Integration and using superscript and subscript for the time range. Needs better notation.
l.173 jth should be j^{th} (superscript).
Table 2. TransFlow has lower EPE on Matched-clean.
l. 244 Abaltions -> Ablations
l. 306 "it significantly increases during training." fragment, please complete the sentence.
[33] should be updated to the peer-reviewed reference.

**Limitations:**

The paper contains a limitation section that raises two relevant limitations of the proposed method: the large memory usage for training and the lack of temporal connections between batches.

---

> ### Author Rebuttal · Authors · 2024-08-07
>
> **Q1: The relationship between image size and memory requirements**
>
> Thank you for your question. During inference, the memory is modest, as shown in Line 305. Moreover, the memory usage could be further reduced via packages like ``flash-attention``. We have compared StreamFlow with the recent method MemFlow and find that they share similar memory usage.
> (MemFlow : Ours = 1: 1.03)
>
> **Q2: Detailed results on the Spring benchmark.**
>
> Thank you for your suggestions. We would add the results in supplements. Results could be seen in the attached PDF.
>
> **Q3: More efficiency analysis.**
>
> Thank you for your questions. Given that some recent multi-frame methods have not been open-sourced, and their papers lack descriptions of efficiency, as well as details such as network layers and widths, which are sensitive to the results, we have not compared them. We have recently updated methods including MemFlow, as shown in the attached PDF. Notably, StreamFlow, MemFlow, and VideoFlow can all adjust the timing by changing the number of refinement iterations, sacrificing accuracy for speed. Here we set iterations to 15 for all of these methods by default.
>
> **Q4: Concern on overfitting.**
>
> Thank you for your question. In fact, StreamFlow has excellent benchmark test and 0-shot test results in multiple datasets, which demonstrates its good generation ability. Its performance in the training set may be related to the size of the dataset, since Sintel and KITTI only contain 1k+ and 200 samples, respectively. StreamFlow shows leading 0-shot performance on Sintel, KITTI, and the larger Spring dataset.
>
> **Q5: Typos.**
>
> We sincerely thank you for pointing out these issues. In the revised version, we will correct each point and carefully proofread the text again.

---

> > ### Comment · Reviewer_ba6H · 2024-08-09
> > **Rebuttal**
> >
> > I thank the authors for their rebuttal and the additional results on the Spring dataset. I also would like to acknowledge the new graph giving an overview and comparison of the efficiency of various methods. But unfortunately, I find that the rebuttal only answers some of my questions.
> >
> > My Q1 was about the relationship between image size and memory, while the authors provided helpful information, the question has not been answered.
> >
> > My Q2 was a request to report all evaluation parameters for MPI-Sintel.
> >
> > In weaknesses, I had pointed out that VideoFlow MOF outperforms StreamFlow for accuracy on Kitti, and that MemFlow has reported superior results. While I understand that different experimental setting lead to different results, the strong claims made in the paper about the relative accuracy of StreamFlow seem overstated.

---

> ### Author Response · Authors · 2024-08-10
> **Response from Authors**
>
> Dear Reviewer,
>
>
> Thank you very much for your valuable and prompt feedback. We sincerely appreciate the time and effort you have invested in reviewing our work and highlighting areas that require further clarification. We are eager to address your concerns as follows:
>
> **Q1: Relationship between image size and memory usage.**
>
> A: With PyTorch 2.2 and flash-attention, using 12 refinements and 4 frames, the GPU memory usage for StreamFlow is as follows. Specifically, when the image size increases by 4 times, the GPU memory usage increases to nearly 4.4 times. When the image size is increased by 9 times, GPU memory usage grows to nearly 20 times. For most current scenarios, StreamFlow still maintains a relatively moderate use of GPU memory. The variation in GPU memory usage may be influenced by underlying optimizations in the framework. We believe the memory usage could be further optimized in the future.
>
>
>
> | Image Size | GPU Memory |
> | ------------ | ------------ |
> | 360x640    | 1.19 G     |
> | 720x1280   | 5.20 G     |
> | 1080x1920 | 24.11 G|
>
> **Q2: All evaluation parameters for MPI-Sintel.**
>
> A: We have collected all measures including ``all/matched/unmatched EPE, d0-10, d10-60, d60-140, s0-10, s10-40, s40+`` in Table 2 of the attched PDF. Compared to its two-frame baseline, StreamFlow performs exceptionally well in unmatched areas, validating its effectiveness in addressing occlusion issues. We would add a more detailed analysis for each metric in the supplements.
>
> **Q3: Relative accuracy of StreamFlow.**
>
> A: Thank you for your insightful comments and for pointing out concerns about the descriptions in this work. Our initial intention is to highlight its notable 0-shot cross-dataset generalization results, and we hope it did not come across as overstating its performance. We will refine the expression in the revised version to minimize potential misunderstandings. For instance, we will revise the content between Lines #223 and #230 to: ``From Table 1, we could learn that StreamFlow achieves excellent 0-shot cross-dataset performance on Sintel and KITTI. Compared to previous methods, StreamFlow reduces the 0-shot end-point error by 0.16 and 0.08 on the challenging Sintel clean and final passes, respectively. On KITTI, StreamFlow surpasses the previous 0-shot results with 0.11 and 17.65% lower EPE and Fl-all metrics. Besides, without self-supervised pre-training or bi-directional flows, StreamFlow attains commendable accuracy and efficiency on the challenging Sintel and KITTI test benchmarks using (C)+T+S+K+H schedule.``
>
> Besides, StreamFlow indeed uses experimental settings different from others. However, this may also highlight StreamFlow’s advantages. For instance: (1) On the Spring dataset, StreamFlow was only trained for 180k iterations while outperforming MemFlow, which was trained for 400k iterations with the same batch size and learning rate. (2) For Sintel testing, StreamFlow was trained for 300k iterations on FlyingThings and 180k on T+S+H+K, while MemFlow was initially trained for 120k (FlyingChairs) and 150k (FlyingThings) in a 2-frame setting, followed by 600k iterations on FlyingThings and 600k on T+S+H+K. Despite this, StreamFlow still delivers good results, especially on Spring. (3) As for VideoFlow, it employs more frames and bidirectional flows for training, achieving excellent results on the KITTI dataset. We will update the result in Table 1 and add the related discussion. However, VideoFlow explores the accuracy and efficiency under bidirectional flow estimation, which differs from the focus of StreamFlow on a non-overlapping, continuous unidirectional flow pipeline. Additionally, its latency is significantly higher than that of StreamFlow. Therefore, A more relevant comparison for highlighting the issues that StreamFlow addresses is with the baseline method, Twins-SKFlow.
>
> In the end, we greatly appreciate your valuable comments and the opportunity to clarify these points. Please kindly let us know if you have any further questions or require additional clarification. We highly value your insights and stand ready to provide any further information that could be helpful.

---

### Official Review · Reviewer_zwXh · 2024-07-10

**Soundness:** 2
**Presentation:** 1
**Contribution:** 2
**Rating:** 4
**Confidence:** 3

**Summary:**

This work focuses on the task of multi-frame optical flow estimation. It challenges the traditional pair-wise flow estimation approach in multi-frame scenarios, which involves redundant calculations. To address this issue, a new framework is proposed that takes multiple frames as input and predicts successive unidirectional flows in a single forward pass.

**Strengths:**

- The proposed method achieves state-of-the-art performance.
- The authors have conducted lots of experiments in the ablation study to justify their designs.

**Weaknesses:**

- The paper is challenging to follow due to several presentation issues.
Specifically, the `Integration’ operation used in Eq. 2 is not clearly defined. Additionally, from Eq. 5 to Eq. 9, using mathematical notations rather than component names to represent the model components in GTR would improve clarity and conciseness. Furthermore, the main figure requires more detailed captions to enhance comprehensibility.

- Lack of further evaluations.
To enhance the validation of the proposed method, the author should also test their model’s generalizability on the widely used Spring [1] benchmark, as it is a standard in many recent optical flow estimation studies.

- Unclear Core Motivation.
In my understanding, the core motivation of this paper is to enable the `simultaneous prediction of successive unidirectional flows in a single forward pass`, as opposed to making per-pair estimations in multi-frame flow estimation scenarios. However, I observed that there are numerous pair-wise operations among the inputs, such as cost volume calculations, in the `single forward pass`.
I suggest that the authors allocate less space to describing these pair-wise operations and instead focus more on discussing and analyzing their designs that enhance efficiency in the `single forward pass`. For example, providing a detailed analysis of the time complexity of the proposed method compared to other methods would be beneficial.
Additionally, it would be helpful for the authors to include comparisons and discussions regarding recent multi-frame flow estimation methods, such as MemFlow [2].


[1] Spring: A High-Resolution High-Detail Dataset and Benchmark for Scene Flow, Optical Flow and Stereo.

[2] MemFlow: Optical Flow Estimation and Prediction with Memory.

**Questions:**

Please see the "Weaknesses" section for my questions and suggestions. If the author can address these concerns, I would be willing to consider raising my rating.

**Limitations:**

Yes.

---

> ### Author Rebuttal · Authors · 2024-08-07
>
> **Q1: Eq.2, Eq. 5~9 could be presented with more clarity and Fig. 3 could be given more captions.**
>
> Thank you for your suggestions. We will revise the formulas for clarity and provide additional captions for Fig. 3 as recommended. We provided a rather general explanation of Integration at Line 169. We will emphasize this expression and provide a more detailed description. Additionally, we will include pseudocode in the supplementary materials for this operation, and release all code and checkpoints once the paper is accepted.
>
> **Q2: Results on the Spring benchmark.**
>
> Thank you for your advice. Whether it was the zero-shot test conducted on the Spring dataset immediately after training on Sintel, or the fine-tuning on Spring, StreamFlow achieved significantly superior overall results and outperformed MemFlow on multiple metrics. It is important to note that StreamFlow was trained for only 180k iterations, which is considerably fewer than the 400k iterations used by MemFlow. Please refer to the attached PDF for details.
>
> **Q3: The redudancy in cost volume and discussions with other recent method such as MemFlow could be included.**
>
> Thank you for your question. In StreamFlow, the cost volume computation is limited to adjacent frames, avoiding redundancy. For instance, with input frames [$I_1$, $I_2$, $I_3$, $I_4$], only correlations [$C_{1,2}, C_{2,3}, C_{3,4}$] are computed for once to derive the flow [$f_{1,2}, f_{2,3}, f_{3,4}$]. The cross-frame information is fused prior to the cost volume calculation via the non-overlapping CSC modules, which is a deliberate design choice in StreamFlow, which explores whether good temporal modeling can still be achieved using a non-overlapping approach.
>
> The work MemFlow (CVPR 24) was posted on arXiv in April 2024, so we had not included it in our comparison previously. It employs the pairwise method and the issue it explores do not overlap with the focus of our work. We have now included a comparison with MemFlow in the attached PDF. StreamFlow demonstrates comparable accuracy with superior latency performance.

---

> ### Author Response · Authors · 2024-08-12
>
> Dear Reviewer,
>
> Thank you once again for taking the time to review our manuscript. We have tried our best to address the questions you raised (please see our responses in the top-level comment and above) and have revised the paper according to the suggestions provided by all reviewers.
>
> Please kindly let us know if there are any additional questions requiring further clarification. Your feedback is highly valued, and we are more than willing to provide any further information that may be helpful.

---

### Official Review · Reviewer_PByw · 2024-07-12

**Soundness:** 3
**Presentation:** 3
**Contribution:** 3
**Rating:** 5
**Confidence:** 4

**Summary:**

The paper presents StreamFlow, a new optical flow estimation method tailored for video inputs. StreamFlow differentiates itself from earlier methods by incorporating a streamlined in-batch multi-frame pipeline that reduces duplicate computations across consecutive frames, thereby enhancing efficiency. Additionally, the introduction of an ISC module and a GTR decoder allows for effective leverage of spatio-temporal information without an increase in parameter count. Extensive experiments demonstrate that StreamFlow surpasses existing methods on the Sintel and KITTI datasets in both accuracy and efficiency.

**Strengths:**

- The method designs are technically sound and well-motivated.
- Quantitative results on the Sintel and KITTI datasets surpass previous works, demonstrating the proposed method's superior generalization capability.
- The proposed methods achieve faster runtime with a parameter-efficient architecture.
- Extensive ablation studies clearly demonstrate the impact of each proposed module and explore multiple variations.

**Weaknesses:**

- The literature review in Section 2 appears insufficient. It should be expanded to include more detailed discussions of related works such as SKFlow and Videoflow, upon which the architectural designs of this paper are based.
- As I understand it, the primary motivation of this research is to design a non-overlapping inference pipeline for multi-frame estimation. This raises a question about how the non-overlapping inference affects accuracy in terms of frame distance, a topic that seems to be omitted from the paper. For instance, when given three input frames [t-1, t, t+1], a pair-wise pipeline might estimate flow f_t,t+1, and then f_t+1,t+2 using [t, t+1, t+2]. In contrast, if the SIM pipeline computes f_t,t+1 using [t-1, t, t+1], it then predicts f_t+1,t+2 using [t+1, t+2, t+3], thereby missing the opportunity to utilize frame t in estimating the flow for f_t+1, t+2. I would expect this loss of information may become more severe with longer frame lengths. Such analysis should be included in the first row of Table 3 (ablation of SIM Pipeline), yet the ablation study was conducted naively, focusing solely on latency without considering factors that affect accuracy.

**Questions:**

Please check the weaknesses above.

**Limitations:**

The authors described limitations adequately.

---

> ### Author Rebuttal · Authors · 2024-08-07
>
> **Q1: More discussions of related works such as SKFlow and VideoFlow could be included in Section 2.**
>
> A: Thank you for your advice. We will add a more detailed discussion to the literature review in Sec 2. For instance, "To address the issue of occlusion，SKFlow begins by  expanding the spatial receptive field and designs effective large convolution kernel modules in the decoder of the flow network, without adding significant computational cost.  VideoFlow, on the other hand, approaches the problem from a temporal perspective, employing TROF and MOP modules to utilize multi-frame temporal cues and bidirectional optical flow to effectively mitigate occlusion issues. StreamFlow also starts from the temporal dimension.
> Differently, it introduces a new non-overlapping pipeline and explores various temporal modeling strategies to be effective in such settings. It addresses the redundancy problems previous pairwise methods including VideoFlow encountered, and achieves excellent accuracy with latency similar to some two-frame methods".
>
> **Q2: Table 3 should include the analysis of the loss of information.**
>
> A: Thank you for your suggestions. We will include an analysis of this in the revised version. To hierarchically display the results of progressively adding various modules during ablation, we did not include the temporal modeling module in the first part of Table 3 previously, and only changed the pipeline. This allowed a direct comparison with results after adding "Tem. modules" in the second part.
>
> We thank you again for the issue you have mentioned, which does indeed exist. As the frame distance increases, the information provided may decrease, as confirmed by the results in the attached PDF.  However, this impact might not necessarily grow larger with more frames, which could be due to the affected frames mainly being distributed at the head or the tail of a group. We could define the longest frame distance that provides effective information as $m$. As the length of the group increases, there will be more frames in the middle of the group (i.e., frames $I_t$ within the interval [$I_{t-m}, I_{t+m}$] all lying within the group), and fewer frames distributed on both ends. As shown in the experiments, the impact was weakened with 4 frames. We will include related discussions in the revised paper, and maybe the future study on an appropriate choice of $m$ is helpful for future multi-frame optical flow work.

---

> ### Author Response · Authors · 2024-08-12
>
> Dear Reviewer,
>
> Thank you once again for taking the time to review our manuscript. We have tried our best to address the questions you raised (please see our responses in the top-level comment and above) and have revised the paper according to the suggestions provided by all reviewers.
>
> Please kindly let us know if there are any additional questions requiring further clarification. Your feedback is highly valued, and we are more than willing to provide any further information that may be helpful.

---

> > ### Comment · Reviewer_PByw · 2024-08-13
> >
> > I thank the authors for their rebuttal and the additional results provided. While there is no compelling reason to oppose publication, and I acknowledge the contributions of the paper, I do not find them to be particularly significant. Therefore, I will maintain my current rating.

---

### Author Rebuttal · Authors · 2024-08-07

Dear Reviewers,

Please refer to the attached one-page PDF that summarizes the added experimental results, which include:

**1. Results on the Spring dataset (CVPR' 23), and the comparison with the recent method MemFlow (CVPR' 24, on arxiv 2404)**

StreamFlow has achieved superior performance on the Spring dataset, surpassing MemFlow. It is important to note that due to limited time during the rebuttal phase, StreamFlow was only trained for 180,000 step instead of 400,000 steps used in MemFlow.

**2. More comparison to other methods on latency.**

Please refer to Figure 1 in the PDF.

**3. Influence of the frame distance.**

Please refer to Table 3 in the PDF.

**4. Detailed results on the MPI-Sintel dataset.**

Please refer to Table 2 in the PDF.


We would like to express our gratitude to all reviewers for providing constructive feedback, which has significantly contributed to the improvement of our paper. We have been working diligently on improving the manuscript in response to your critiques. Please see our reviewer-specific feedback for more detailed information.

---

### Decision · Program_Chairs · 2024-09-25

**Decision:**

Accept (poster)

**Comment:**

The paper presents a method for optical flow estimation in videos that overcomes the need for repetitive overlapping computation in multi-frame optical flow. All the reviewers appreciate the results as state-of-the-art. Evaluations requested by zwXh on the Spring benchmark are provided in the rebuttal and support the effectiveness of the proposed method. The motivation of the proposed method is also clarified in the rebuttal noting that the cost volume computation avoids redundancy. The strength of results are also noted by PByw, whose request for discussion with respect to SKFlow and VideoFlow are addressed by the rebuttal. While ba6H is supportive of acceptance, additional results are requested and sufficiently provided by the rebuttal. Overall, the AC agrees with the majority opinion that the paper is well-motivated, proposes a novel approach and presents strong results. The authors are encouraged to release code and checkpoints as indicated by the rebuttal to zwXh, while also making the improvements to presentation listed there. In balance, the paper may be accepted for NeurIPS.